# Broader impacts of an intervention to transform school environments on student behaviour and school functioning: post hoc analyses from the INCLUSIVE cluster randomised controlled trial

Christopher Bonell [1], Matthew Dodd,[1] Elizabeth Allen,[2] Leonardo Bevilacqua,[3] Jennifer McGowan,[3] Charles Opondo,[2,4] Joanna Sturgess,[2,5] Diana Elbourne,[2,6] Emily Warren,[2,7] Russell M Viner [8,9]

For numbered affiliations see end of article.

**Correspondence to**
Dr Christopher Bonell;
chris.bonell@lshtm.ac.uk

## ABSTRACT

**Background** We have previously reported benefits for reduced bullying, smoking, alcohol and other drug use and mental health from a trial of 'Learning Together', an intervention that aimed to modify school environments and implement restorative practice and a social and emotional skill curriculum.

**Objectives** To conduct post hoc theory-driven analyses of broader impacts.

**Design** Cluster randomised trial.

**Settings** 40 state secondary schools in southern England.

**Participants** Students aged 11/12 years at baseline.

**Outcomes** Student self-reported measures at 24 and 36 months of: cyberbullying victimisation and perpetration; observations of other students perpetrating aggressive behaviours at school; own perpetration of aggressive behaviours in and outside school; perceived lack of safety at school; participation in school disciplinary procedures; truancy and e-cigarette use.

**Results** We found evidence of multiple impacts on other health (reduced e-cigarette use, cyberbullying perpetration, perpetration of aggressive behaviours) and educational (reduced participation in school disciplinary procedures and truancy) outcomes.

**Conclusion** These analyses suggested that the intervention was effective in bringing about a broader range of beneficial outcomes, adding to the evidence that the intervention is a promising approach to promote adolescent health via an intervention that is attractive to schools.

**Trial registration number** ISRCTN10751359.

## INTRODUCTION

We have previously reported the results of our main trial analyses from the INCLUSIVE cluster randomised controlled trial (RCT) of a multicomponent intervention aiming to transform school environments to render these healthier places, reporting effectiveness

### Strengths and limitations of this study

► This study was a rigorously conducted experimental evaluation.

► These are additional analyses that were not included in our original protocol, so caution is required in the interpretation of significant findings.

► However, the analyses are guided by explicit, theory-driven hypotheses, as set out in our introduction, rather than being the product of subjecting all measures to analysis and merely reporting significant findings.

► We relied on student self-reports as these were less likely than school routine data to be biased by delivery of the intervention.

► While our measure of student participation in disciplinary procedures had high inter-item reliability, this was lower for our measure of student perpetration of aggressive behaviours in or outside school, so our conclusions regarding effects on this outcome should be cautious.

across multiple health domains.[1] In this paper, we aimed to explore the extent to which the intervention 'disrupted' the school 'system' to achieve more impacts.

There is increasing interest in the implications of 'systems' thinking for evaluating health interventions.[2] Interventions can be viewed as 'interruptions' to complex systems, the consequences of which may go beyond the primary and secondary health outcomes assessed by trials. This perspective is of particular relevance for our intervention since this aimed to promote students' health not by improving individual students' knowledge, skills or attitudes, but rather by modifying the overall

school environment so that it is more engaging and thus an easier environment to choose healthy rather than risky behaviours. Due to our interventions' focus on settings and use of multiple components, it is particularly likely that its impact might have gone beyond our prehypothesised primary and secondary outcomes to impact on the school systems' broader functioning.[3] This paper therefore draws on theory to develop and test hypotheses about what the broader impacts of our intervention might have been.

Our intervention aimed to transform the school environment to make this more salutogenic, informed by previous evidence of effective interventions.[4–6] It aimed to do so by: (a) using 'restorative approaches' to address conflict; (b) rendering schools more participative by involving students and staff in an action group to review local data on student experiences and use this to change school policies related to behaviour management, and lead the intervention and (c) providing a social and emotional skills curriculum for students aged 12–15 years.[1 7] Restorative approaches aim to enable victims to communicate to perpetrators the harms experienced, and enable perpetrators to recognise and take steps to remedy this and avoid further harms.[8] Restorative approaches include primary prevention of conflict (via 'circle-time', which brings students together to build and maintain relationships) and/or secondary prevention to resolve incidents (such as 'conferencing' to address serious incidents).

The intervention was supported by a theory of change that was informed by an appropriate settings-based social theory: the theory of human functioning and school organisation.[9] We theorised that schools can reduce bullying and aggression by transforming the school environment to build student commitment to learning and sense of belonging in school. It was theorised that this in turn can be achieved by improving relationships between and among staff and students (via the action group and restorative practice) and by better integrating students' academic education and broader personal development (via the curriculum and restorative practice). It was further theorised that by increasing student commitment to and belonging in school, this would reduce student interest and involvement in antisocial peer groups and behaviours.

Our main trial paper examined intervention effects on the primary and secondary outcomes described in the trial protocol. We reported a range of significant intervention effects in terms of reduced bullying victimisation (co-primary outcome) and use of tobacco, alcohol and other drugs, reduced contact with police, as well as improved mental well-being, psychological functioning and health-related quality of life among adolescent students (secondary outcomes) at 36 months (table 1).[1] We found no significant effect for perpetration of aggression in school (co-primary outcome) or for age of sexual debut, use of contraception at first sex, bullying perpetration or use of National Health Service (secondary outcomes). The intervention was implemented with variable fidelity, with this being lower in year 3. Training, action groups and restorative practices but not the curriculum were delivered with good fidelity.[1]

**Table 1** Outcomes assessed in the main trial paper and the post hoc analysis

| Outcomes assessed in main trial paper | Evidence of significant beneficial effect on this outcome | Outcomes assessed in this post hoc analysis |
|---|---|---|
| Bullying victimisation | √ | Cyberbullying victimisation |
| Aggression perpetration in school | | Cyberbullying perpetration |
| Health-related quality of life | √ | Observing other students' perpetrating aggression in school |
| Mental well-being | √ | Aggression perpetration either in or out of school |
| Psychological problems | √ | Perceived lack of safety at school |
| Bullying perpetration | | Participation in school discipline procedures |
| Cigarette smoking | √ | Truancy |
| Alcohol use | √ | E-cigarette use |
| Drunkenness | √ | |
| Illicit drug use | √ | |
| Age of sexual debut | | |
| Contraception at first sex | | |
| NHS service use | | |
| Police contact | √ | |

NHS, National Health Service.

The first area where we anticipated beneficial broader impacts is cyberbullying. Our main trial analyses reported effects of the intervention on our primary outcome of reduced bullying victimisation (Gatehouse Bullying Scale).[10] This is an important result given the prevalence of bullying[11] and its association with concurrent and future physical and mental health harms.[12–18] However, this analysis was insensitive to any effects of the intervention on cyberbullying. Cyberbullying is an increasingly prevalent aspect of bullying, associated with significant harms.[19] We did not include this in our list of primary or secondary outcomes because this mostly occurs outside school. However, assessing this would be appropriate given that our intervention is theorised to work by decreasing student interest in antisocial behaviour in general not limited to the school site.

Our second area of exploration is perpetration of aggression. Our main trial analyses found no evidence of effects on our other primary outcome of self-reported perpetration of school-based aggression (Edinburgh Study of Youth Transitions (ESYTC) measure).[20] This was an unexpected finding

given the reduction in bullying victimisation, and given the intervention was theorised to reduce bullying victimisation and perpetration of aggression via a common mechanism involving increased student commitment to school and reducing student involvement in antisocial peer groups and behaviours. A systematic review has previously concluded that trials of whole-school interventions addressing violence sometimes find effects on victimisation but not perpetration, possibly because participants under-report perpetration of socially unacceptable behaviours particularly in school.[5] Our student questionnaire also included a measure for students to report their observations of other students perpetrating aggressive behaviours, thus perhaps providing a broader assessment of aggressive behaviours in school and less prone to under-reporting. Therefore, we hypothesised that we will find effects of the intervention on this measure of student-reported observations of other students perpetrating aggressive behaviours at school. Furthermore, our primary measure of perpetration of aggression focused only on school-based behaviours. Since our intervention aimed to reduce students' general involvement in antisocial peer groups and behaviours, rather than merely reducing such behaviours in school, we hypothesised that the intervention would be effective in reducing a broader measure of students' own perpetration of aggressive behaviours not specific to school which we included in our questionnaire.

Our third focus for this paper is on impacts of the intervention on the overall functioning of the school system. Interventions effective in reducing bullying and promoting student health are more likely to be scaled up if schools and policymakers can see evidence that such interventions also reduce school workloads and enhance education.[21] Our theory of change centred on enhancing student commitment to school and reducing student involvement in antisocial behaviours, and we found effects not only for reduced bullying victimisation but also for increased student commitment to school.[22] We therefore hypothesised that this will translate into students reporting: feeling safer at school, less participation in school disciplinary procedures and less truancy.

Our final focus in this paper is on e-cigarette use. As indicated earlier, we found effects for smoking but our measure focused on the smoking of tobacco rather than use of e-cigarettes. However, we would also expect the intervention to reduce the latter. There are increasing concerns about the increasing prevalence of e-cigarette use among young people with some evidence that this is associated with subsequent increase in smoking tobacco.[23] We therefore hypothesised that rates of use of e-cigarettes are lower among schools in the intervention group.

In summary, we hypothesised that the intervention was effective not only with regard to the primary and secondary outcomes measured described in our protocol, but also in promoting a broader range of unintended but beneficial impacts via its disruption of the school system, reducing student-reported: cyberbullying victimisation and perpetration; observations of other students perpetrating aggressive behaviours at school; own perpetration of aggressive

behaviours in and outside school; perceived lack of safety at school; participation in school disciplinary procedures; truancy and e-cigarette use (table 1). Given our previous finding that intervention effects on primary and secondary outcomes were apparent at 36-month but not 24-month follow-up, which is in line with previous evidence that the effects of whole-school interventions build over time as they take time to transform the school environment,[6] we hypothesised that this would also apply to the outcomes examined in this paper.

## METHODS
Full details of the intervention and trial were reported in our protocol and main trial report.[1 7] We conducted a two-arm parallel repeat cross-sectional cluster RCT of the intervention in 40 secondary schools in south-east England. To be included, schools had government inspections rating of 'requires improvement' or above and were recruited by the trial team via emails. Our student population consisted of all students: at baseline in 2014 who were at the end of year 7 (11–12 years); who were then in year 9 at interim 24-month follow-up in 2016 and who were in year 10 at final 36-month follow-up in 2017. Some students moved schools, hence the study was repeated cross-sectional since all were included in analyses. Students were surveyed using paper questionnaires in classes under exam conditions by trained fieldworkers blinded to allocation. After baseline surveys, schools were allocated 1:1 to intervention or control by computer-generated random numbers stratified by: single-sex versus mixed-sex school; school-level student free-school-meal eligibility (0%–23%; >23%) indicating poverty and General Certificate of Secondary Education results accounting for student baseline attainment (above/below the median score of 1000 for England).

The intervention involved all staff in intervention schools receiving training to use restorative practice to prevent and address student conflicts. Approximately 5–10 key staff per school were trained in-depth to deliver restorative conferences dealing with more serious incidents. All schools received a manual to guide the convening and running of a school action group comprising at least six staff and six students, led by a member of the school's senior leadership team. An external facilitator supported action groups in the first two but not the third year of intervention, when they moved to being self-directed. Action groups reviewed anonymised findings from the school's baseline survey to understand local needs and aimed to coordinate the intervention and revise policies so that these supported the use of restorative practice. Schools were provided with materials to guide delivery of a social and emotional skills curriculum for students in years 8–10 to receive 5–10 hours teaching per year. The curriculum addressed bullying and aggression but not specific to a particular setting such as school or online. Schools in the control group continued with usual practice.

Our measures analysed in this paper are described in table 2. Each of these were included in student

| Table 2 | Outcome measures | | | |
|---|---|---|---|---|
| **Outcome measure** | **Question** | **Responses** | **Source** | **Variable** |
| Cyberbullying victimisation | Have you been bullied through mobile phone use or on the internet in the last 3 months? | No I haven't<br>Yes, once or twice<br>Yes, two or three times a month<br>Yes, about once a week<br>Yes, several times a week or more | Adapted from Daphne measure of cyberbullying | Binary any yes/no |
| Cyberbullying perpetration | Have you ever bullied anyone else using your mobile phone or using the internet? | No I haven't<br>Yes, once or twice<br>Yes, two or three times a month<br>Yes, about once a week<br>Yes, several times a week or more | Adapted from Daphne measure of cyberbullying | Binary any yes/no |
| Student-reported observations of other students perpetrating aggressive behaviours at school | Which of the following have you seen happen at this school in the last 3 months of school | Boys fighting<br>Girls fighting<br>Someone threatening someone<br>A student trying to hurt another student<br>Someone robbing money or a mobile phone<br>Someone letting off a firework<br>Someone carrying a knife | New | Score out of 7 (point per item) |
| Perpetration of antisocial behaviour in or outside school | During the last 3 months of school<br><br>Did you ever carry a knife or other weapon with you for protection or in case it was needed in a fight?<br>Did you use force, threats or a weapon to steal money or something else from somebody?<br>Did you damage or destroy property that did not belong to you on purpose (eg, windows, cars or street lights)?<br>Did you ever set fire or try to set fire to something on purpose (eg, bus shelter, shop, etc)? | No<br><br><br><br>Yes | Adapted from ESYTC measure of antisocial behaviour | Score out of 4 (4 items with each no/yes 0–1) |

**Table 2** Continued

| Outcome measure | Question | Responses | Source | Variable |
|---|---|---|---|---|
| Participation in school disciplinary procedures | During the last 3 months of school how often did these things happen to you because of something you had done wrong? | | | |
| | The school got in touch with my parents by letter or telephone about an incident | 0 times | ESYTC measure of school discipline | Score out of 18 (6 items with each scored 0–3) |
| | I got a punishment and my parents were informed about that | 1 or 2 times | | |
| | I was given detention | 3 or 4 times | | |
| | I was sent to the head of year, deputy head or head teacher for my behaviour | 5 or more times | | |
| | I was put on a conduct/ behaviour sheet | | | |
| | I was given extra homework to do | | | |
| Truancy | During the last 3 months of school have you skipped/bunked off school? | No / Yes | Ripple measure of truancy | Binary yes/no |
| E-cigarette use | Which of the following best describes you? | I currently smoke e-cigarettes | New | Binary—ever/never |
| | | I have tried e-cigarettes in the past 12 months but do not currently smoke them | | |
| | | I have tried e-cigarettes longer than 12 months ago but do not currently smoke them | | |
| | | I have never tried e-cigarettes | | |
| Perceived lack of school safety | Do you feel safe at this school? | All of the time | HSE measure of school safety | Binary— some of the time/never versus other options |
| | | Most of the time/some of the time | | |
| | | Never | | |

ESYTC, Edinburgh Study of Youth Transitions; HSE, Healthy School Ethos.

questionnaires used to assess trial outcomes but did not form part of our specified trial outcomes. We adapted Smith and colleagues' measure of cyberbullying perpetration and victimisation.[24] We developed a new single-item measure of student-reported observations of other students perpetrating aggressive behaviours at school, where students indicated which behaviours they had observed at school to provide a quantitative measure scored 0–7. We examined students' own perpetration of aggressive behaviours in or outside school using a modified four-item version of the ESYTC measure of antisocial behaviours.[20] Students reported which behaviours they had engaged in to give a quantitative score 0–4. We assessed perceived lack of school safety using a single item derived from the Healthy School Ethos study.[25]

We assessed student participation in school disciplinary procedures using the six-item ESYTC measure of school discipline.[20] This assessed students' frequency of engagement (never; one or two times; three or four times and five or more times) with six disciplinary procedures to provide a quantitative score 0–18. We assessed school truancy using a student-reported single-item measure previously used in the Ripple trial.[26] We developed a new single-item measure of e-cigarette use.

## PATIENT AND PUBLIC INVOLVEMENT

The trial involved young people from The National Children's Bureau Young Researchers' Group in advising on intervention and research methods during three meetings

at the set-up phase. School action groups comprised part of the intervention and enabled students to participate in planning and coordinating intervention activities.

As with our analysis of primary and secondary outcomes, our analyses of outcomes in this paper were intention-to-treat, including all schools and participants at each wave. Each outcome measure was analysed using a separate mixed model with the measure from each time point treated as a repeated measure. Fixed effects of time (baseline, 24 months and 36 months) and the interaction between arm and time were specified, and estimated baseline measures were constrained to be identical in the two arms of the trial. This is equivalent to adjusting for baseline but enables data from all participants to contribute to the analysis, even where there are missing data at follow-up. We specified random effects for school and participants, to allow for correlations within schools and repeated measures within participants. We undertook analyses adjusted for baseline measures of outcomes, sex, ethnicity, socioeconomic status (Family Affluence Scale (FAS)) as well as for the school-stratifying factors.

We used appropriate multilevel models to examine the effects of the intervention. For quantitative measures, we used mixed linear-regression models with random effects at the level of participants and schools to estimate adjusted mean differences (MD) between arms. For binary outcomes, we used mixed-effects logistic regression models, with random effects for schools and individuals, reporting unadjusted and adjusted OR. Evidence for any moderation of intervention effects on our outcomes by student sex and socioeconomic status (FAS) was assessed by Wald tests for the treatment by subgroup interaction terms. We also calculated Cronbach's alpha to assess the inter-item reliability of our multiquestion measures of student perpetration of aggressive behaviours in or outside school and student participation in school disciplinary procedures.

Informed consent was sought from head teachers for randomisation and intervention, and from students, deemed competent by schools to do so, for participation in surveys. Parents were informed about the study and could withdraw their children from research activities.

## RESULTS
In total, 6667 students in 40 schools provided data at baseline, representing a participation rate of 93.6% of registered students (92.9% in intervention arm; 94.3% control arm). Student characteristics and baseline reports of the outcomes examined in this paper are reported in table 3, with good balance between arms.

All schools participated in the surveys at 24 and 36 months, with student participation rates being similar by arms (figure 1).

Cronbach's alpha for our ESYTC measures of student perpetration of aggressive behaviours in or outside school was 0.55 and for our ESYTC measure of student participation in school disciplinary procedures was 0.79.

Our broader student outcomes at 24 and 36 months are reported in table 4. At 24 months comparing intervention with control schools, we found lower rates of: cyberbullying victimisation (OR=0.77; 95% CI 0.61, 0.98; p=0.035) but not perpetration and e-cigarette use (OR=0.60 95% CI 0.43, 0.83; p=0.002). Students in intervention schools were more likely to report lack of perceived school safety at 24 months than controls (OR=1.38, 95% CI 1.10, 1.75; p=0.006). There was no evidence of difference between arms in: student-reported observations of other students perpetrating aggressive behaviours at school; perpetration of aggressive behaviours in or outside school or truancy. There was weak to moderate evidence of lower reported participation in school disciplinary procedures in intervention compared with control schools at 24 months (MD=−0.16, 95% CI −0.32, 0.00; p=0.043).

At 36 months comparing intervention and control schools, we found reduced rates of: cyberbullying perpetration (OR=0.65, 95% CI 0.48, 0.88; p=0.005) but not victimisation; perpetration of aggression in or outside school (MD=−0.031, 95% CI −0.056 to 0.006; p=0.016); participation in school disciplinary procedures (MD=−0.320, 95% CI −0.480 to 0.150; p<0.001); truancy (OR=0.64, 95% CI 0.49, 0.84; p=0.001) and e-cigarette use (OR=0.59, 95% CI 0.42, 0.82; p=0.002). There was weak to moderate evidence of lower student-reported observations of other students perpetrating aggressive behaviours at school (MD=0.10, 95% CI 0.00, 0.20; p=0.049). There were no evidence of difference in perceived school safety.

Table 5 presents outcomes and follow-up points for which there was evidence of moderation. We found evidence that intervention effects on cyberbullying perpetration at 24 months were moderated by student sex, such that effects were larger for boys (p=0.002). Intervention effects on observed aggression by other students at 24 months were moderated by student sex and socioeconomic status, with effects larger for girls (p=0.02) and affluent students (p=0.03). Effects on perceived lack of school safety at 24 months were larger for girls (p=0.001), and at 36 months were larger for students from poorer families (p=0.002). Effects on participation in school disciplinary procedures at 24 and 36 months were larger for boys (p<0.001 and 0.001, respectively). Effects on truancy at 24 months were larger for boys (p=0.015). Effects on e-cigarette use at 24 and 36 months were larger for boys (p=0.014 and <0.001 respectively).

## DISCUSSION
### Summary of key findings
We reported an analysis of broader system impacts on student health and school functioning outcomes of this settings-based intervention aim to render schools more health-promoting environments which was previously reported to be effective in reducing bullying victimisation and use of alcohol, tobacco and drugs, as well as promoting mental and physical health.[1] We found evidence at 36 months but not 24 months of intervention

**Table 3** Characteristics of schools and students at baseline by trial arm

| School characteristics | Control 20 schools | Intervention 20 schools | Overall 40 schools |
|---|---|---|---|
| School sex mix, n (%) | | | |
| Mixed | 15 (75.0) | 15 (75.0) | 30 (75.0) |
| Girls | 3 (15.0) | 4 (20.0) | 7 (17.5) |
| Boys | 2 (10.0) | 1 (5.0) | 3 (7.5) |
| Ofsted rating*, n (%) | | | |
| Excellent | 5 (25.0) | 6 (30.0) | 11 (27.5) |
| Good | 13 (65.0) | 12 (60.0) | 25 (62.5) |
| Requires improvement | 2 (10.0) | 2 (10.0) | 4 (10.0) |
| Value added score, mean (SD) | 1003 (24.8) | 1004 (20.4) | 1003 (22.4) |
| Proportion of students on free school means, mean (SD) | 36 (18.0) | 35 (22.0) | 36 (20.0) |
| IDACI, mean (SD) | 0.26 (0.2) | 0.24 (0.2) | 0.25 (0.2) |
| Student socio-demographic characteristics | 3347 students† | 3320 students† | 6667 students† |
| Age, mean (SD) | 12 (0.4) | 12 (0.4) | 12 (0.4) |
| Sex, n (%) | | | |
| Male | 1639 (49.9) | 1464 (44.9) | 3103 (47.3) |
| Female | 1649 (50.2) | 1804 (55.2) | 3453 (52.7) |
| Ethnicity, n (%) | | | |
| White British | 1391 (41.5) | 1221 (37.3) | 2612 (39.7) |
| White other | 291 (8.8) | 273 (8.3) | 564 (8.6) |
| Asian/Asian British | 859 (25.9) | 786 (24.0) | 1645 (25.0) |
| Black/Black British | 384 (11.6) | 535 (16.4) | 919 (14.0) |
| Chinese/Chinese British | 11 (0.3) | 35 (1.1) | 46 (0.7) |
| Mixed ethnicity | 238 (7.2) | 224 (6.9) | 462 (7.0) |
| Other | 140 (4.2) | 198 (6.1) | 338 (5.1) |
| Family affluence scale, mean (SD) | 6 (1.8) | 6 (1.8) | 6 (1.8) |
| Student baseline rates of outcomes | | | |
| Cyberbullying perpetration, n (%) | 290 (8.9) | 279 (8.6) | 569 (8.7) |
| Cyberbullying victimisation, n (%) | 522 (16.0) | 467 (14.5) | 989 (15.3) |
| Truancy, n (%) | 189 (5.9) | 182 (5.8) | 371 (5.8) |
| E-cigarette use, n (%) | 187 (5.8) | 131 (4.2) | 318 (5.0) |
| Perceived lack of school safety, n (%) | 493 (15.6) | 440 (14.5) | 933 (15.1) |
| Student-reported observations of other students perpetrating aggressive behaviours at school, mean (SD) | 2.30 (1.61) | 2.04 (1.65) | 2.17 (1.64) |
| Perpetration of aggressive behaviour in/outside school, mean (SD) | 0.06 (0.31) | 0.06 (0.33) | 0.06 (0.32) |
| Participation in school disciplinary procedures, mean (SD) | 2.47 (2.96) | 2.39 (3.00) | 2.43 (2.98) |

*One control school did not have an Ofsted rating.
†The number of students who responded at this survey; actual number of responses to each question varies, but item non-response is similar across arms.
ADACI, Income Deprivation Affecting Children Index.

effects on: cyberbullying perpetration; student observations of aggression by other students; students' own perpetration of aggressive behaviours in or outside school; truancy and participation in school disciplinary procedures. There was evidence of an effect on increased student perceptions of lack of school safety at 24 months but not 36 months. There was also evidence of an effect at 24 months but not 36 months on cyberbullying victimisation. We found evidence of an effect on e-cigarette use at both time points.

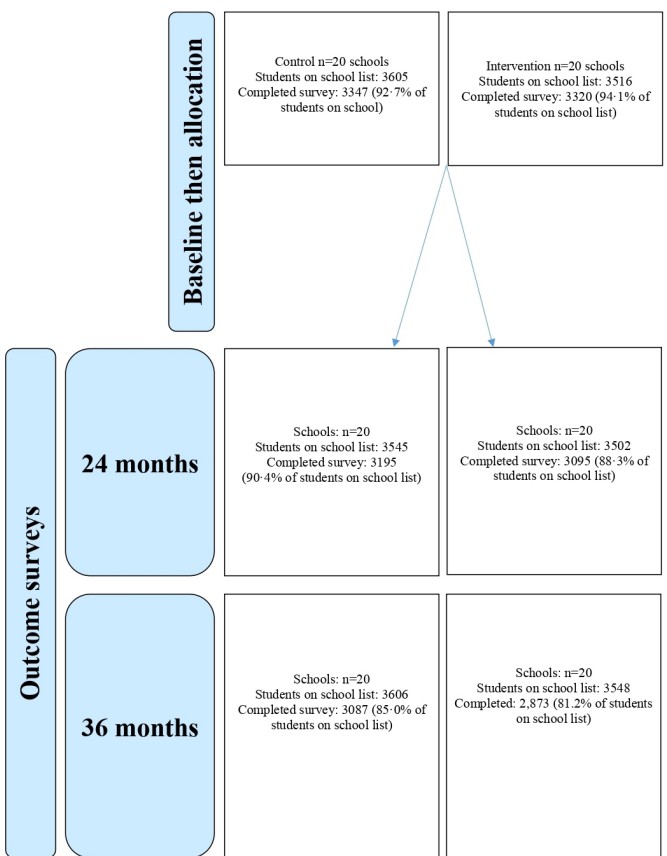

**Figure 1** Trial participants.

These findings suggest that our intervention, which aimed to reduce bullying via making schools more engaging environments, and which did not explicitly focus on cyberbullying, might nonetheless have been effective in reducing this. Our findings also suggest that the intervention might, contrary to the main analyses, have reduced rates of aggression including aggression beyond the school environment. However, results across time points and measures are somewhat inconsistent, probably as a result of chance. Our intervention also appears to have reduced student use of e-cigarettes, which is an important finding given increasing concerns about this as a gateway to tobacco use.[23] This evidence of additional health impacts provides further evidence in support of the intervention theory of change that it is possible to improve young people's health across a range of areas by addressing the school as a potentially salutogenic environment rather than merely as a setting for individual-focused health education in classrooms. Furthermore, our finding of broader impacts on school functioning in terms of reduced truancy and student involvement in discipline systems provides evidence of the knock-on consequences of a health intervention disrupting the school system to achieve impacts on the domain of education.

Intervention effects were moderated in some cases by student sex and family affluence. Effects were larger for boys regarding reduced cyberbullying perpetration

and truancy at 24 months, and reduced participation in school disciplinary procedures and e-cigarette use at 24 and 36 months. For girls, intervention effects were greater regarding reduced observed aggression by other students and decreased perceived school safety at 24 months. Effects for decreased perceived school safety at 36 months were also larger for students from poorer families. Effects on reduced observed aggression at 24 months were larger for affluent students. These findings contrast with moderator analyses for our primary and secondary outcomes, where benefits were generally larger for boys and no different for those from poorer families.[1]

Our finding of increased student perceptions of lack of safety in intervention versus control schools at 24 months, and among poorer students at 36 months, is of concern. We noted that this association runs counter to our previous findings for actual rates of bullying victimisation[1] and to the findings in this paper on aggression. This finding may be due to chance. However, it might be explained by the intervention's focus on bullying and aggression sensitising students to issues of safety, leading them to feel unsafe at the initial (24 month) follow-up but dissipating as the intervention became normalised and exerted positive effects on bullying and aggression.

These moderator analyses add to the evidence from our main analyses that the intervention might generally have been more effective for boys than girls. As with our main trial analyses, there was less evidence for moderation by socioeconomic status. These findings of gender inequity of effects is in line with some previous research suggesting that whole-school interventions, including those to reduce violence, can sometimes be more effective for boys than girls.[1 4] This might be because, in such interventions, violence and other problem behaviours among boys receive more attention than those experienced by girls.

## Strengths and limitations

This study was a rigorously conducted experimental evaluation. These are additional analyses not included in our original protocol. Therefore, caution is required in the interpretation of significant findings. However, the analyses are guided by explicit, theory-driven hypotheses, as set out in our introduction, rather than being the product of subjecting all measures to analysis and merely reporting significant findings. We relied on student self-reports as these were less likely than school routine data to be biased by delivery of the intervention. Where possible, we used reliable existing measures. While our measure of student participation in disciplinary procedures had high inter-item reliability, this was lower for our measure of student perpetration of aggressive behaviours in or outside school so our conclusions regarding effects on this outcome should be cautious.

## Implications for policy and research

Our findings suggest that the intervention disrupted school systems to achieve a range of unintended but beneficial

**Table 4** Outcomes at 24-month and 36-month follow-up

| Outcomes | 24 months | | | | 36 months | | | |
|---|---|---|---|---|---|---|---|---|
| | Control n/N (%) or mean (SD), N | Intervention n/N (%) or mean (SD), N (%) | Unadjusted OR or MD (95% CI), p value | Adjusted OR or MD (95% CI), p value | Control n/N (%) or mean (SD), N | Intervention n/N (%) or mean (SD), N (%) | Unadjusted OR or MD (95% CI), p value | Adjusted OR or MD (95% CI), p value |
| Cyberbullying victimisation | 443/3116 (14.2%) | 340/2993 (11.4%) | OR=0.79 (0.63 to 0.99), 0.041 | OR=0.77 (0.61 to 0.98), 0.035 | 347/2987 (11.6%) | 266/2754 (9.7%) | OR=0.83 (0.65 to 1.05), 0.121 | OR=0.80 (0.62 to 1.05), 0.110 |
| Cyberbullying perpetration | 257/3112 (8.3%) | 229/2984 (7.7%) | OR=0.93 (0.71 to 1.21), 0.572 | OR=0.90 (0.67 to 1.19), 0.450 | 287/3008 (9.5%) | 193/2766 (7.0%) | OR=0.66 (0.51 to 0.87), 0.003 | OR=0.65 (0.48 to 0.88), 0.005 |
| Perceived lack of school safety | 543/3106 (17.5%) | 568/2944 (7.7%) | OR=1.39 (1.11 to 1.74), 0.004 | OR=1.39 (1.10 to 1.75), 0.006 | 601/2919 (20.6%) | 532/2682 (19.8%) | OR=1.08 (0.86 to 1.36), 0.495 | OR=1.05 (0.83 to 1.34), 0.440 |
| Student-reported observations of other students perpetrating aggressive behaviours at school | 2.43 (1.65), 3179 | 2.12 (1.71), 3063 | MD=−0.05 (−0.14 to 0.05), 0.344 | MD=−0.08 (−0.18 to 0.01), 0.096 | 2.23 (1.77), 3069 | 2.07 (1.76), 2834 | MD=0.11 (0.01 to 0.20), 0.030 | MD=0.10 (0.00 to 0.20), 0.049 |
| Perpetration of aggressive behaviours in/outside school | 0.10 (0.44), 3149 | 0.08 (0.39), 3017 | MD=−0.014 (−0.037 to 0.009), 0.229 | MD=−0.009 (−0.034 to 0.015), 0.456 | 0.12 (0.50), 3023 | 0.09 (0.43), 2778 | MD=−0.030 (−0.054 to −0.007), 0.012 | MD=−0.031 (−0.056 to −0.006), 0.016 |
| Participation in school disciplinary procedures | 2.74 (3.28), 3128 | 2.36 (3.12), 3009 | MD=−0.222 (−0.375 to −0.069), 0.004 | MD=−0.160 (−0.320 to 0), 0.043 | 2.55 (3.21), 3017 | 2.15 (2.94), 2757 | MD=−0.326 (−0.484 to −0.169), <0.001 | MD=−0.320 (−0.480 to −0.150), <0.001 |
| Truancy | 340/3088 (11.0%) | 299/2961 (10.1%) | OR=0.93 (0.72 to 1.20), 0.582 | OR=0.92 (0.70 to 1.21), 0.551 | 426/2981 (14.3%) | 294/2750 (10.7%) | OR=0.66 (0.51 to 0.85), 0.001 | OR=0.64 (0.49 to 0.84), 0.001 |
| E-cigarette use | 558/3061 (18.2%) | 352/2913 (12.1%) | OR=0.62 (0.45 to 0.86), 0.004 | OR=0.60 (0.43 to 0.83), 0.002 | 630/2946 (21.4%) | 394/2692 (14.6%) | OR=0.60 (0.44 to 0.83), 0.002 | OR=0.59 (0.42 to 0.82), 0.002 |

**Table 5** Moderation

| Outcome | Follow-up (months) where evidence of moderation | Moderator Variable | Categories | Association (95% CI) | Interaction (p value) |
|---|---|---|---|---|---|
| Perpetration cyberbullying | 24 | Sex | Boys | OR=0.61 (0.41 to 0.89) | 0.002 |
| | | | Girls | 1.19 (0.85 to 1.67) | |
| Observed aggression | 24 | Sex | Boys | MD=0.01 (−0.11 to 0.13) | 0.02 |
| | | | Girls | MD=−0.15 (−0.26 to −0.04) | |
| | | Family affluence | Low | MD=−0.18 (−0.62 to 0.25) | 0.03 |
| | | | Middle | MD=0.04 (−0.09 to 0.18) | |
| | | | High | MD=−0.14 (−0.25 to −−0.03) | |
| Lack of safety | 24 | Sex | Boys | OR=0.99 (0.73 to 1.35) | 0.001 |
| | | | Girls | OR=1.74 (1.33 to 2.27) | |
| | 36 | Family affluence | Low | OR=3.07 (0.99 to 9.54) | 0.002 |
| | | | Middle | OR=0.72 (0.51 to 1.01) | |
| | | | High | OR=1.23 (0.94 to 1.60) | |
| Participation in school disciplinary procedures | 24 | Sex | Boys | MD=−0.39 (−0.60 to −0.19) | <0.001 |
| | | | Girls | MD=0.02 (−0.17 to 0.20) | |
| | 36 | Sex | Boys | MD=−0.55 (−0.76 to −0.34) | 0.001 |
| | | | Girls | MD=−0.14 (−0.33 to 0.05) | |
| Truancy | 24 | Sex | Boys | OR=0.69 (0.48 to 0.99) | 0.015 |
| | | | Girls | OR=1.13 (0.83 to 1.55) | |
| E-cigarette use | 24 | Sex | Boys | OR=0.45 (0.30 to 0.67) | 0.014 |
| | | | Girls | OR=0.80 (0.54 to 1.20 | |
| | 36 | | Boys | OR=0.35 (0.23 to 0.53 | <0.001 |
| | | | Girls | OR=0.94 (0.63 to 1.40). | |

impacts on student health and school functioning not captured in the main trial analyses.[2 3] These results suggest that it is possible to achieve public health improvements across a range of outcomes using a single coordinated intervention which focuses on environmental transformation rather than individual behaviour change. This is important given the impracticality of implementing different interventions for multiple outcomes in schools.[21] Our findings also suggest important benefits for education and school functioning. It appears that the intervention's previously reported effects on reducing bullying victimisation and improving student commitment to school[1 22] translated into reduced student truancy and participation in school disciplinary procedures. This is especially an important evidence for school leaders suggesting the potential educational benefits of whole-school health interventions. Our analyses suggest this intervention worked more effectively for boys than girls. Further research is needed on how to ensure school-based interventions are more equitable, perhaps by ensuring these address less overt forms of student disengagement and conflict.

**Author affiliations**
[1]Public Health and Policy, London School of Hygiene and Tropical Medicine, London, UK

[2]Department of Medical Statistics, London School of Hygiene & Tropical Medicine, London, UK
[3]University College London, London, UK
[4]London School of Hygiene and Tropical Medicine, London, UK
[5]Department of Population Health, London School of Hygiene and Tropical Medicine, London, UK
[6]EPH, LSHTM, London, UK
[7]Faculty of Epidemiology and Population Health, London School of Hygiene & Tropical Medicine, London, UK
[8]Population, Policy and Practice Research Programme, UCL Institute of Child Health, London, UK
[9]Institute of Child Health, University College London, Londn, UK

**Acknowledgements** We are grateful to the staff and students of participating schools for their dedication to the intervention and completion of the outcome surveys and process evaluation surveys and interviews. We are very grateful for the advice and support of our Trial Steering Committee and Data Monitoring Committee. All those who contributed significantly to this work are listed in these acknowledgements.

**Contributors** All those who contributed significantly to this work are listed as authors. CB and RMV directed the trial from which the data are drawn. CB, MD, EA and CO designed the analysis for this paper. MD implemented this design and undertook the analysis. DE provided additional statistical expertise. CB wrote the first draft of the paper which was then edited and commented on by MD, EA, LB, JM, CO, JS, DE, EW and RMV.

**Funding** This work was supported by the National Institute for Health Research (NIHR) in England under its Public Health Research Board (12/153/60) and the Education Endowment Foundation (no grant number). The views expressed in this publication are those of the authors and do not necessarily reflect those of the

National Health Service, the NIHR or the Department of Health for England. The study funders (NIHR and Education Endowment Foundation) played no role in the study; collection, analysis and interpretation of data; the writing of the report or the decision to submit the manuscript for publication.

**Competing interests**  None declared.

**Patient and public involvement**  Patients and/or the public were involved in the design, or conduct, or reporting, or dissemination plans of this research. Refer to the Methods section for further details.

**Patient consent for publication**  Not required.

**Ethics approval**  The trial was prospectively registered as ISRCTN10751359 with the ISRCTN Registry on 30 January 2014 and approved by the UCL Ethics Committee (ref 5248/001).

**Provenance and peer review**  Not commissioned; externally peer reviewed.

**Data availability statement**  Data are available upon reasonable request. Data are available upon reasonable request.

**ORCID iDs**
Christopher Bonell http://orcid.org/0000-0002-6253-6498
Russell M Viner http://orcid.org/0000-0003-3047-2247

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
