## [Reviewer comments · BMJ Open]

ARTICLE DETAILS

TITLE (PROVISIONAL)	Broader impacts of an intervention to transform school environments on student behaviour and school functioning: post hoc analyses from the INCLUSIVE cluster randomised controlled trial
AUTHORS	Bonell, Christopher; Dodd, Matthew; Allen, Elizabeth; Bevilacqua, Leonardo; McGowan, Jennifer; Opondo, Charles; Sturgess, Joanna; Elbourne, Diana; Warren, Emily; Viner, Russell

VERSION 1 – REVIEW

REVIEWER	Sharyn Burns Curtin University, Western Australian Centre for Health Promotion Research, School of Public Health
REVIEW RETURNED	29-Aug-2019

GENERAL COMMENTS	A well-developed paper that provides encouraging findings around the longer term impact of school-based interventions. The authors have stated that not all measures were included in this paper. They have described the intervention. It is important to discuss does of intervention. For example, part of the intervention included 5-10 hours of curriculum. Was this measured? Were there differences between the time spent delivering curriculum? Similarly how effectively were the restorative practices implemented? While it is likely to be beyond the scope of this paper to discuss these measures it would be useful to include some reference (perhaps in the limitations; or reference to other publications if these findings have been reported elsewhere). Minor edits: Line 37: edit 'over school environment so that these ...' .. change to 'so that it ...' Use alcohol and other drugs (as opposed to alcohol and drugs) Page 10, Line 26 Capital The Page 20, Line 58 – check sentence structure
--

REVIEWER	Lucy Bowes University of Oxford, United Kingdom
REVIEW RETURNED	22-Sep-2019

GENERAL COMMENTS	This well-written manuscript describes additional analyses from the impressive INCLUSIVE cluster randomised trial of school-based intervention targeting bullying and health promotion. The main paper has already been published, showing positive results across primary and secondary outcomes. This manuscript presents additional analyses based on the premise that the intervention may have disrupted the school system in ways that impact on more broader outcomes relating to health (e.g. e-
---

	cigarette use), behaviour (e.g. truancy and perpetration of cyber-bullying) and perception of school safety. Of note it was good to see the authors make explicit that "analyses are guided by explicit, theory-driven hypotheses...rather than being the product of subjecting all measures to analysis and merely reporting significant findings". I have some minor suggestions, outlined below, which I believe could further strengthen this already impressive manuscript.  1. As mentioned above, it is great that these analyses were theory driven; however although one can check the original protocol to see what additional analyses (beyond the main paper) could have been done, it would help to have a little more information in this manuscript. I wonder whether a table could help here - e.g. with a list of primary and secondary measures in one column, a column showing whether they were tested in the original study (and possibly some simple way of indicating whether the effect was significant) or not, and a third column showing what was used in the present study, to help the reader get a sense of what has and hasn't been analysed. 2. The results section (aside from Table 3) focuses on the significance of the results; it would be useful for these results to be put in context, so that the relative effect sizes are made more clear to the reader in the written results. 3. Please amend the sentence on page 20, third paragraph: "Effects for decreased perceived school safety at 36 months were also larger for students from poorer families" to "Effects for decreased perceived lack of school safety at 36 months were also larger for students from poorer families" - assuming that was what was meant here, and that this wasn't an observation of increased harm in this group! 4. Were the moderation effects observed (broadly) in line with the main trial results? It would be useful to reference this in the discussion section. 5. I think it would be better see the unadjusted and adjusted outcome results - perhaps dividing Table 3 into two tables (24 months and 36 months) if needed. It was very interesting to see in the main study that the unadjusted and adjusted results were very similar (suggesting minimal effects of the covariates included), and I wonder if the same is true for the current study. 6. I am a fan of keeping decimal places for p values the same though out - and suggest that two is good (more than that suggests a level of precision that is not really true..). The above are all pretty minor points - overall I very much enjoyed reading this manuscript and felt it makes another important addition to the field.
--	---

REVIEWER	Obioha Ukoumuhne NIHR CLAHRC South West Peninsula (PenARC), University of Exeter
REVIEW RETURNED	18-Oct-2019

GENERAL COMMENTS	The paper is generally very clearly written. I have some minor comments/thoughts:
---

1) The study is described at the beginning of the Methods section as a “repeated cross-sectional” cluster RCT. I thought this was potentially misleading because this term is usually used in studies where the same schools are followed up but the participating pupils are different at each wave. So initially I thought that the baseline assessment was performed on Year 7 pupils in 2016 with the follow-up assessments on completed different pupils who were in Year 7 in 2016 (for the 24 month follow-up) and in Year 7 in 2017 (for the 36 month follow-up). It is only subsequently that it becomes clearer that the study attempts to follow a cohort of children from Year 7 through to Year 10 but that some children leave the cohort and some presumably join it. So to me this is closer to a cohort cluster RCT than a repeated cross-sectional cluster RCT, although it does have elements of both. Perhaps worth adding a sentence or two about this aspect of the design near the beginning of the Methods. Related to this, in the 5th line of the Methods section when referring to the 24 and 36 month follow-ups it might be worth adding in brackets the calendar year and the school Year group of assessment to make it clearer that some kind of cohort is being followed.

2) In the statistical analysis section it is indicated that for the main analysis the data were analysed in longitudinal form (so that there are up to 3 records per participating child) and that mixed models were fitted using the predictor variables trial arm status, time and the interaction between trial arm status and time. The authors say that the “estimated baseline measures were constrained to be identical in the two arms of the trial” but it is my understanding that to do this the model should include only the time and interaction variables as predictors and NOT trial arms status. I say this based on the following papers:

Coffman CJ, Edelman D, Woolson RF. To condition or not condition? Analysing ‘change’ in longitudinal randomised controlled trials. *BMJ Open* 2016;6:e013096.
doi:10.1136/bmjopen-2016-013096

Hooper R and colleagues. Analysis of cluster randomised trials with an assessment of outcome at baseline. *BMJ* 2018; 360: k1121

3) Were mixed logistic regression models fitted for the analysis of binary outcomes? The statistical analysis section only states that odds ratios were reported.

4) When examining subgroup effects I assume these analyses were not carried out using longitudinal data as otherwise there would be a three-way interaction between trial arm status, time and the moderator?

5) In Table 1 when reporting percentages for the school level characteristics I'd be inclined to round these to whole numbers as there are only 20 schools in each trial arm. It might also be worth deleting the third column that presents summaries for all 40 schools.

6) In the heading row of Table 3 the authors have written “..or mean (SD), N (%)” when they only need “..or mean (SD), N”. The heading for the Intervention arm at 36 months needs further details. A percentage is missing for the Intervention arm at 24

	month follow up for the “Cyber-bullying perpetration” variable. It might make Table 3 easier to read if the table is converted to landscape orientation and adding 4 separate columns to indicate the sample size for each combination of trial arm status and follow-up. 7) A couple of times the paper refers to results as being of “borderline statistical significance” where I think a term like “weak to moderate evidence of a true effect” might be better and give less emphasis to the 0.05 threshold p-value. 8) In Table 4 add a footnote to indicate what “association” statistic is reported or better still write in the cells “MD=0.61” (for mean difference) or “OR=0.61” (for odds ratio) so it is clear what statistic is reported for each outcome. Might be worth changing the column heading “p-value” to “interaction p-value”? 9) In the second paragraph of the discussion section (line 8) change “provide” to “provideS”. 10) I wonder if it would be possible to modify the title slightly to give some idea of the types of outcomes that are examined. I guess the broad area are aspects of child behaviour and school functioning? 11) I think there is a word missing in the sentence in the Introduction section (near the end of 6th paragraph) that begins “However, given that our intervention is theorised to work...”. 12) In the second paragraph of the Method section change “Action groups reviewed anonymised finding...” to “Action groups reviewed anonymised findingS...”. 13) The scoring range for 1 or 2 variables that were analysed as quantitative outcomes seemed more ordinal (i.e., on a 0 to 4 scale) than continuous but I guess the sample size is more than large enough to make inferences valid. 14) First sentence in “Patient and Public Involvement” needs a capital “t”.
--	---

VERSION 1 – AUTHOR RESPONSE

Reviewer 1	
It is important to discuss dose of intervention. For example, part of the intervention included 5-10 hours of curriculum. Was this measured? Were there differences between the time spent delivering curriculum? Similarly how effectively were the restorative practices implemented? While it is likely to be beyond the scope of this paper to discuss these measures it would be useful to include some reference (perhaps in the limitations; or reference to other publications if these findings have been reported elsewhere).	As the reviewer suggests it is beyond the scope of this paper to present detailed evidence on intervention dose and fidelity. This evidence has already been published in our main trial paper. However, in the introduction of the current paper we briefly refer to this evidence to put the new evidence in context.

Line 37: edit 'over school environment so that these ...' .. change to 'so that it ...'	We have corrected this.
Page 20, Line 58 – check sentence structure	We have corrected this.
Use alcohol and other drugs (as opposed to alcohol and drugs)	We have corrected this.
Page 10, Line 26 Capital The	We have corrected this.
Reviewer 2	
As mentioned above, it is great that these analyses were theory driven; however although one can check the original protocol to see what additional analyses (beyond the main paper) could have been done, it would help to have a little more information in this manuscript. I wonder whether a table could help here - e.g. with a list of primary and secondary measures in one column, a column showing whether they were tested in the original study (and possible some simple way of indicating whether the effect was significant) or not, and a third column showing what was used in the present study, to help the reader get a sense of what has and hasn't been analysed.	The main trial paper examined the primary and secondary outcomes described in our protocol. We now clarify this in the introduction section. Here we are also clearer which primary and secondary outcomes were not found to have been significantly affected in the analyses reported in the main paper. The present paper reports on other outcomes not described in the protocol – this point is clarified at the end of the introduction. These are described in table 1.
The results section (aside from Table 3) focuses on the significance of the results; it would be useful for these results to be put in context, so that the relative effect sizes are made more clear to the reader in the written results.	The results have been amended to report effect sizes.
Please amend the sentence on page 20, third paragraph: "Effects for decreased perceived school safety at 36 months were also larger for students from poorer families" to "Effects for decreased perceived lack of school safety at 36 months were also larger for students from poorer families" - assuming that was what was meant here, and that this wasn't an observation of increased harm in this group!	The sentence is correct. We have amended our discussion section, which already discusses this finding at 24 months to encompass the finding for some students at 36 months.
Were the moderation effects observed (broadly) in line with the main trial results? It would be useful to reference this in the discussion section.	We have added a line to our discussion to compare with the main trial results for moderation.
I think it would be better see the unadjusted and adjusted outcome results - perhaps dividing Table 3 into two tables (24 months and 36 months) if needed. It was very interesting to see in the main study that the unadjusted and adjusted results were very similar (suggesting minimal effects of the covariates included), and I wonder if the same is true for the current study.	We have added these results into table 3.
I am a fan of keeping decimal places for p values the same though out - and suggest that two is good (more than that suggests a level of precision that is not really true..).	We have amended this so that we consistently report: 1 decimal places for percentages; 2 decimal places for effect sizes and standard deviations; and 3 decimal places for mean differences and P values.
Reviewer 3	
The study is described at the beginning of the Methods section as a "repeated cross-sectional" cluster RCT. I thought this was potentially misleading because this term is usually used in studies where the same schools are followed up but the participating pupils are different at each	We have clarified this in our methods section.

wave. So initially I thought that the baseline assessment was performed on Year 7 pupils in 2016 with the follow-up assessments on completed different pupils who were in Year 7 in 2016 (for the 24 month follow-up) and in Year 7 in 2017 (for the 36 month follow-up). It is only subsequently that it becomes clearer that the study attempts to follow a cohort of children from Year 7 through to Year 10 but that some children leave the cohort and some presumably join it. So to me this is closer to a cohort cluster RCT than a repeated cross-sectional cluster RCT, although it does have elements of both. Perhaps worth adding a sentence or two about this aspect of the design near the beginning of the Methods.	
Related to this, in the 5th line of the Methods section when referring to the 24 and 36 month follow-ups it might be worth adding in brackets the calendar year and the school Year group of assessment to make it clearer that some kind of cohort is being followed.	This has been added.
In the statistical analysis section it is indicated that for the main analysis the data were analysed in longitudinal form (so that there are up to 3 records per participating child) and that mixed models were fitted using the predictor variables trial arm status, time and the interaction between trial arm status and time. The authors say that the “estimated baseline measures were constrained to be identical in the two arms of the trial” but it is my understanding that to do this the model should include only the time and interaction variables as predictors and NOT trial arms status. I say this based on the following papers: Coffman CJ, Edelman D, Woolson RF. To condition or not condition? Analysing ‘change’ in longitudinal randomised controlled trials. BMJ Open 2016;6:e013096. doi:10.1136/bmjopen-2016-013096 Hooper R and colleagues. Analysis of cluster randomised trials with an assessment of outcome at baseline. BMJ 2018; 360: k1121	As recommended by the reviewer, our analyses included the predictor variables, time and the interaction between trial arm and time. There was no trial arm indicator in the models. We have deleted this error from the methods section.
Were mixed logistic regression models fitted for the analysis of binary outcomes? The statistical analysis section only states that odds ratios were reported.	Mixed effects logistic regression models were fitted for binary outcomes (and contained random effects for schools and individuals, just like for continuous outcomes). We have edited the methods to reflect this.
When examining subgroup effects, I assume these analyses were not carried out using longitudinal data as otherwise there would be a three-way interaction between trial arm status, time and the moderator?	Subgroup analyses were performed using longitudinal data and included three-way interactions between trial arm status, time and the moderator.

In Table 1 when reporting percentages for the school level characteristics I'd be inclined to round these to whole numbers as there are only 20 schools in each trial arm. It might also be worth deleting the third column that presents summaries for all 40 schools.	We assumed the reviewer is referring to table 2. We have decided to be consistent in reporting only one decimal place for percentages (see above). We would prefer to retain the overall column in table 2 as this might be of use for readers interested in the overall prevalence of some of the baseline outcomes.
In the heading row of Table 3 the authors have written “..or mean (SD), N (%)” when they only need “..or mean (SD), N”.	We have deleted this.
The heading for the Intervention arm at 36 months needs further details.	We have added this.
A percentage is missing for the Intervention arm at 24 month follow up for the “Cyber-bullying perpetration” variable.	We have added this.
It might make Table 3 easier to read if the table is converted to landscape orientation and adding 4 separate columns to indicate the sample size for each combination of trial arm status and follow-up.	We have converted table 3 to a landscape format to report the unadjusted analyses requested by reviewer 2. Given these 2 additional columns, we don't think we can add any other additional columns and so propose to retain the rest of the table format.
A couple of times the paper refers to results as being of “borderline statistical significance” where I think a term like “weak to moderate evidence of a true effect” might be better and give less emphasis to the 0.05 threshold p-value.	We have amended this.
In Table 4 add a footnote to indicate what “association” statistic is reported or better still write in the cells “MD=0.61” (for mean difference) or “OR=0.61” (for odds ratio) so it is clear what statistic is reported for each outcome. Might be worth changing the column heading “p-value” to “interaction p-value”?	We have amended these.
In the second paragraph of the discussion section (line 8) change “provide” to “provideS”.	We have corrected this.
I wonder if it would be possible to modify the title slightly to give some idea of the types of outcomes that are examined. I guess the broad area are aspects of child behaviour and school functioning?	We have added this.
I think there is a word missing in the sentence in the Introduction section (near the end of 6th paragraph) that begins “However, given that our intervention is theorised to work...”.	We have edited this sentence.
In the second paragraph of the Method section change “Action groups reviewed anonymised finding...” to “Action groups reviewed anonymised findingS...”.	We have corrected this.
The scoring range for 1 or 2 variables that were analysed as quantitative outcomes seemed more ordinal (i.e., on a 0 to 4 scale) than continuous but I guess the sample size is more than large enough to make inferences valid.	We thank the reviewer for this comment and have retained the current wording.
First sentence in “Patient and Public Involvement” needs a capital “t”.	We have corrected this.

VERSION 2 – REVIEW

REVIEWER	Obioha Ukoumunne NIHR ARC South West Peninsula (PenARC), University of Exeter
REVIEW RETURNED	09-Nov-2019
GENERAL COMMENTS	I am happy with the authors' responses to my comments.